# Roadside sales activities in a South Pacific Island (Bora-Bora) reveal sustainable strategies for local food supply during a pandemic

Lana Minier[1,2‡], Manon Fourrière[1‡], Emma Gairin[1,3‡], Alannah Gourlaouen[1‡], Stéphanie Krimou[1,4], Cécile Berthe[1], Tehani Maueau[5], Milton Doom[5], Vincent Sturny[2], Suzanne C. Mills[1,4], David Lecchini[1,4‡], Frédéric Bertucci[1,6‡]*

1 PSL Research University: EPHE-UPVD-CNRS, UAR 3278 CRIOBE, Moorea, French Polynesia,
2 Polynésienne des Eaux, Vaitape, Bora-Bora, French Polynesia, 3 Okinawa Institute of Science and
Technology, Okinawa, Japan, 4 Laboratoire d'Excellence "CORAIL", Perpignan, France, 5 Association Ia
Vai Ma Noa Bora-Bora, Bora-Bora, French Polynesia, 6 UMR MARBEC, University of Montpellier, CNRS,
IFREMER, IRD, Sète, France

‡ LM, MF, EG, and AG contributed equally as first authors to this work. DL and FB contributed equally as last
authors to this work.
* frederic.bertucci@ird.fr

**Data Availability Statement:** All data underlying the findings reported in the present study are

## Abstract

During the COVID-19 pandemic, the reduced exports and imports as well as the lack of activity due to the interruption in the international tourism economy seriously impacted food security in many Pacific Islands. People often returned to natural resources to provide for themselves, their families, or to generate income. On Bora-Bora Island, the major tourist destination in French Polynesia, roadside sales are widespread. Our study analyses the impact of the COVID-19 pandemic on roadside sales activities through a census of roadside stalls on the five Bora-Bora districts conducted before (January and February 2020), during (from March 2020 to October 2021) and after (from November to December 2021) health-related activity and travel restrictions. Our results showed that the marketing system for local products (fruits, vegetables, cooked meals, and fish) increased in the form of roadside sales during the COVID-19 in two of the five districts of Bora-Bora. Roadside selling would be an alternative system for providing food to the population at Bora-Bora during a global crisis and that could reveal itself sustainable after this pandemic.

## Introduction

The social and sanitary restrictions put in place at the beginning of COVID-19 pandemic and during the year following its appearance (2021–2022) have severely slowed down the economy of many Pacific islands and impacted the world imports and exports of food products [1, 2]. For example, Bennett et al. [3] showed that the COVID-19 pandemic has had negative consequences on most small-scale fisheries across the Pacific, notably due to shutdowns of fisheries,

provided as a supplementary information file (S1 Table).

**Funding:** This work has received several grants: DL: Fondation de France (2019-08602, https://www.fondationdefrance.org/), ANR-19-CE34-0006-Manini and ANR-19-CE14-0010-SENSO (https://anr.fr/); Office Français de la Biodiversité (AFB/2019/385 – OFB.20.0888, https://www.ofb.gouv.fr/) VS: Polynésienne des Eaux (https://www.polynesienne-des-eaux.pf/) LM: ANRT grant (CIFRE 2021/1268, https://www.anrt.asso.fr/fr). The funders had no role in study design, data collection and analysis, decision to publish, or preparation of the manuscript.

**Competing interests:** The authors have declared that no competing interests exist.

knock-on economic effects from market disruptions, and increased illegal and unregulated fishing. The social restrictions impacted the implementation of Sustainable Development Goals related to income, nutrition, and food security (SDGs 1,2,3) [4, 5]. Although the social lockdowns may have offered some relief to the natural world [6], and notably to marine ecosystems [7–10], the subsequent breakdown of supply chains and fisheries, and reduced employment in tourism, also presented socio-economic challenges, especially to Pacific populations. This may have led in particular to a greater reliance on local resources and harvesting of fish resources [11]. Indeed, international border closures blocked export trade, increasing pressures on local resources to meet food requirements [12–14], and undermining food security in many Pacific communities. Moreover, since the 1950s, many Pacific Islands economies that were based on subsistence agriculture and fishing have transitioned to economies that today rely primarily on tourism [15]. Thus, it soon became apparent that populations in the Pacific required a functional local food system to deal with world economic and social crises, such as that of COVID-19 [16].

The roadside activity of selling land and sea products (fruits, vegetables, cooked meals, fish produced/captured locally and snack bars) in Small Island Developing States (SIDS) is an interesting model for sustainability science [17], which is a recent and fast-growing research area focusing on ensuring that now and in the future the needs of human populations will be met while protecting our environment [18]. Roadside stalls are temporary structures or stands run by producers (fishermen or farmers) to sell their products directly to the local population, most often in locations where a common market-place is not present [19]. In French Polynesia (i.e., Pacific Island Territory), roadside sales mainly consist of agricultural products (fruits, cooked meals, vegetables, and food products) and fish [20]. Many small-scale fishers sell their catch of the day by the road, in front of their houses or near their boats [21]. This type of (often) unofficial market system enables local populations to buy local products and help smallholders. The roadside sale of land and sea products not only provides local food security, but is also an example where supply directly meets demand [20]. The present study aimed to investigate the impact of the COVID-19 pandemic on a customary food system in a SIDS, Bora-Bora, a French Polynesian Island. Goods sold on the roadside provide a way of measuring local food production, may reflect the economic impact of the COVID-19 crisis, and allow for a better understanding of how local inhabitants adapt to partial or total unemployment and lack of tourists due to the social and travel restrictions.

In French Polynesia, there were four periods of social and travel restrictions related to COVID-19: 1) a total lockdown from March to May 2020 (no local and international flight); 2) a ban on foreign tourists coming to Polynesia from February to May 2021 (but there was some local flights from Tahiti and flights from France); 3) a total lockdown from August to mid-September 2021; and 4) a partial lockdown with tourism and fishing activities only during the week from mid-September to October 2021 (international, France and local flights opened). These social and travel restrictions had a strong impact on tourism, fisheries, and recreation: without tourists, hotels temporarily closed, professional fishermen could not sell their fish to hotels, and nautical activities stopped. Bora-Bora Island is one of the major international tourist destinations of French Polynesia, famous for its luxury aqua-centric resorts [22]. Before the COVID-19 pandemic, around 100,000 tourists visited Bora-Bora annually, with over 90% coming from the USA [23]. However, between March 2020 and October 2021, tourism decreased by at least 70% and the hotels were largely empty, which slowed down the island's economy and left numerous workers unemployed or in part-time conditions (data from French Polynesia Tourism Department: https://tahititourisme.fr/).

The present study was opportunistic and aimed to describe the sales activity of land and sea products (fruits, vegetables, cooked meals, and fish produced/captured locally) at Bora-Bora

before (January and February 2020) and during the COVID-19 period (from March 2020 to October 2021) as well as after the end of all restrictions (from November to December 2021). To do so, we made an inventory of the number of roadside stands and identify the types of products that were sold around Bora-Bora Island.

## Methods

On Bora-Bora (as elsewhere in French Polynesia) [20], three types of roadside stalls exist: 1/ Roadside stalls of land products from Monday to Saturday—Most stalls sold land-based products (fruits, vegetables, and sometimes pastries) on weekdays (Monday to Saturday). 2/ Roadside stalls of cooked meals on Sunday—The type of goods were mostly Polynesian pastries (*firi firi* in Tahitian), coconut bread, and lunch boxes (*ma'a*). These stalls sold products on Sunday morning for family lunches and after religious ceremonies. On weekdays, these stalls were absent. 3/ Roadside stalls of fish from Monday to Sunday—Stalls selling fish were recorded every day of the week. Fishermen usually sell their catch of the day directly on the roadside. The fish were sold in groups hanging along a rope (*tui*) or in bags.

The censuses of roadside stalls (each stall sells only one type of product) were conducted along the coastal road, that circles the island, in the five Bora-Bora districts (Vaitape, Faanui, Anau, Matira, and Povai—Fig 1). The road sellers always put stalls in front of their houses or their boat. It is usually rare to move from their house/boat to another site [20]. Thus, to identify each roadside stall and not count the same stall twice, we took the GPS coordinates (S1 Table) and we would also ask the seller if their stand was always in the same location. We did not ask for the name of the person as they often prefer to remain anonymous (even though the island is small and we often already knew the seller).

The study was conducted from January 2020 to December 2021. Our goal was to identify the maximum absolute number of roadside stalls as possible present at any time in Bora-Bora during the three restriction periods: before, during, and after the pandemic-related restrictions. As the presence of a same stall may vary during the day, on different days of the week, and between weeks for various reasons (inefficient fishing, rain limiting fruit harvesting, plant collection period, holidays, etc.), the censuses were conducted randomly at different times of the day, on different days and over different weeks, either all around the island or on a preferential district (depending on the number of stalls). The absence of a stall during a tour of the island did not mean it was considered absent. For each tour of the island, we noted all the stalls present, regardless of the food sold. Once back at the laboratory, we checked if the observed stalls had already been noted in our Excel database. Stalls were surveyed 9 times during the week (Monday to Saturday) and 3 times on Sunday before COVID-19 period; 16 times during the week and 7 times on Sunday during COVID-19 pandemic; and 11 times during the week and 4 times on Sunday after COVID-19 period.

Overall, the data refer only to the absolute total number of roadside sellers present at any time, with one value for each sampling period per village, and not to the average number of vendors observed per day. We calculated the effect sizes of statistical measures in order to compare the number of roadside stalls as the difference between means of data obtained during and after the lockdown compared to before it using the R package "dabestr" [24] with R Studio version 2022.12.0 (R version 4.2.0). Differences were considered significant when the 95% confidence interval (95% CI) of the effect size was different from zero.

## Results

The overall number of roadside stalls in Bora-Bora did not change significantly both during (N = 15, mean difference = 1.73, 95% CI = -3.13–6.67) and after (N = 15, mean

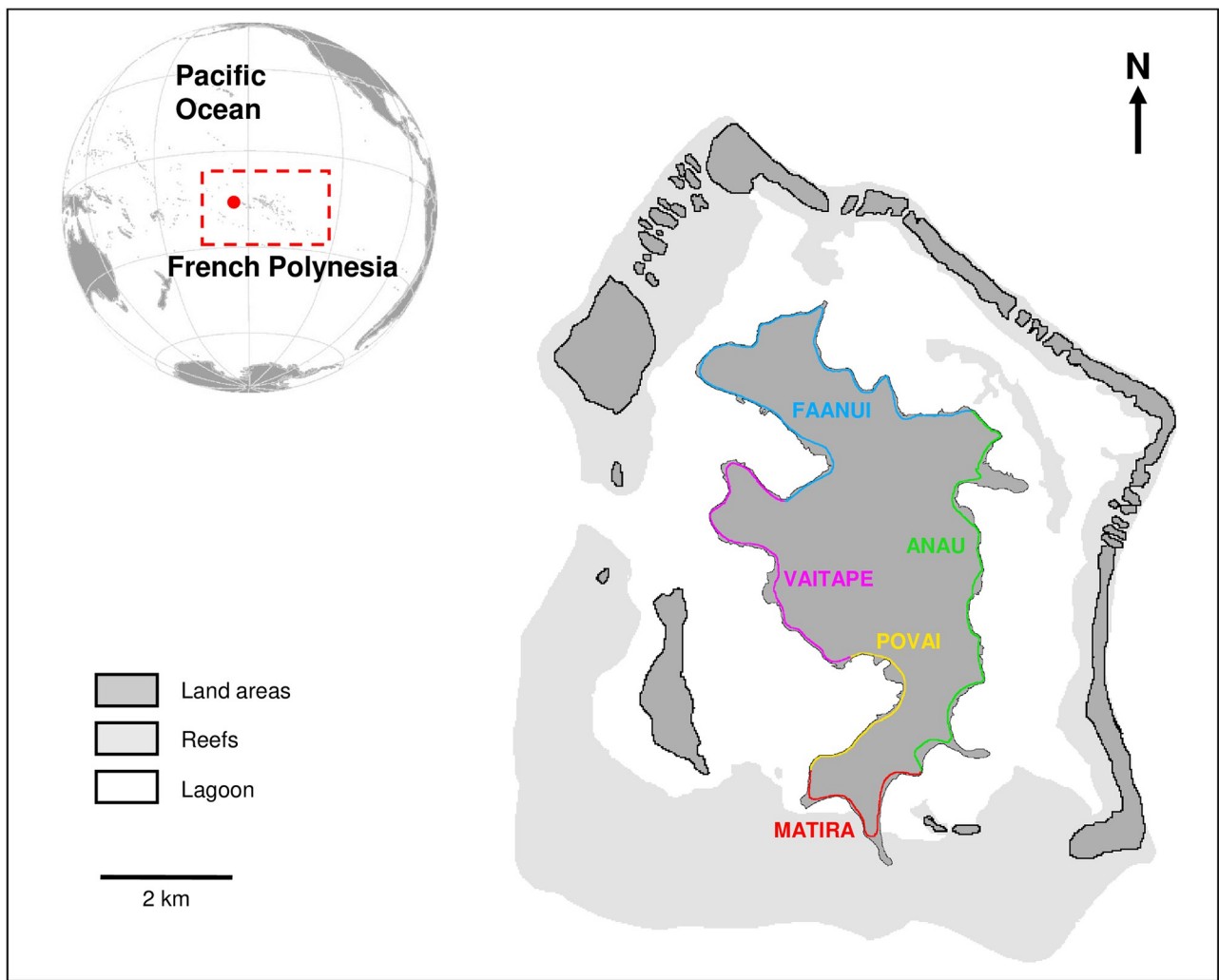

**Fig 1. Map of Bora-Bora showing the main road across the five different districts where roadside stalls were surveyed.** Stands are represented by red dots. Vaitape is the main village on the island and the most touristic part of Bora-Bora along with Povai and Matira. Anau and Faanui are more rural parts of the island. There are no sales of land or sea products on the different islets surrounding the main island of Bora-Bora. The only road is the one that runs along the coast all around the island. Reprinted from an aerial photograph under a CC BY license, with permission from CRIOBE, original copyright 2018.

difference = 1.4, 95% CI = -3.53–7.13) the COVID-19 pandemic compared to before it (Fig 2).

At the scale of Bora-Bora (without distinction according to the districts), the surveys of roadside stalls of land products from Monday to Saturday showed an increase of 45% in the stalls number over the two-years observational periods, with an increase of 36% during the pandemic-related restrictions and an additional 7% increase after the end of the restrictions (Table 1a). However, these changes were not significant (during: N = 5, mean difference = 1.6, 95% CI = -4–9.6; after: N = 5, mean difference = 2, 95% CI = -4.2–12.2). The surveys of roadside stands of cooked meals on Sunday showed an overall 24% increase in the number of Sunday roadside stalls on the island from before to after the restrictions—with a 35% increase during the pandemic restrictions, but an 8% decrease after the restriction period ended (Table 1b). Again, no significant changes were found (during: N = 5, mean difference = 2.6,

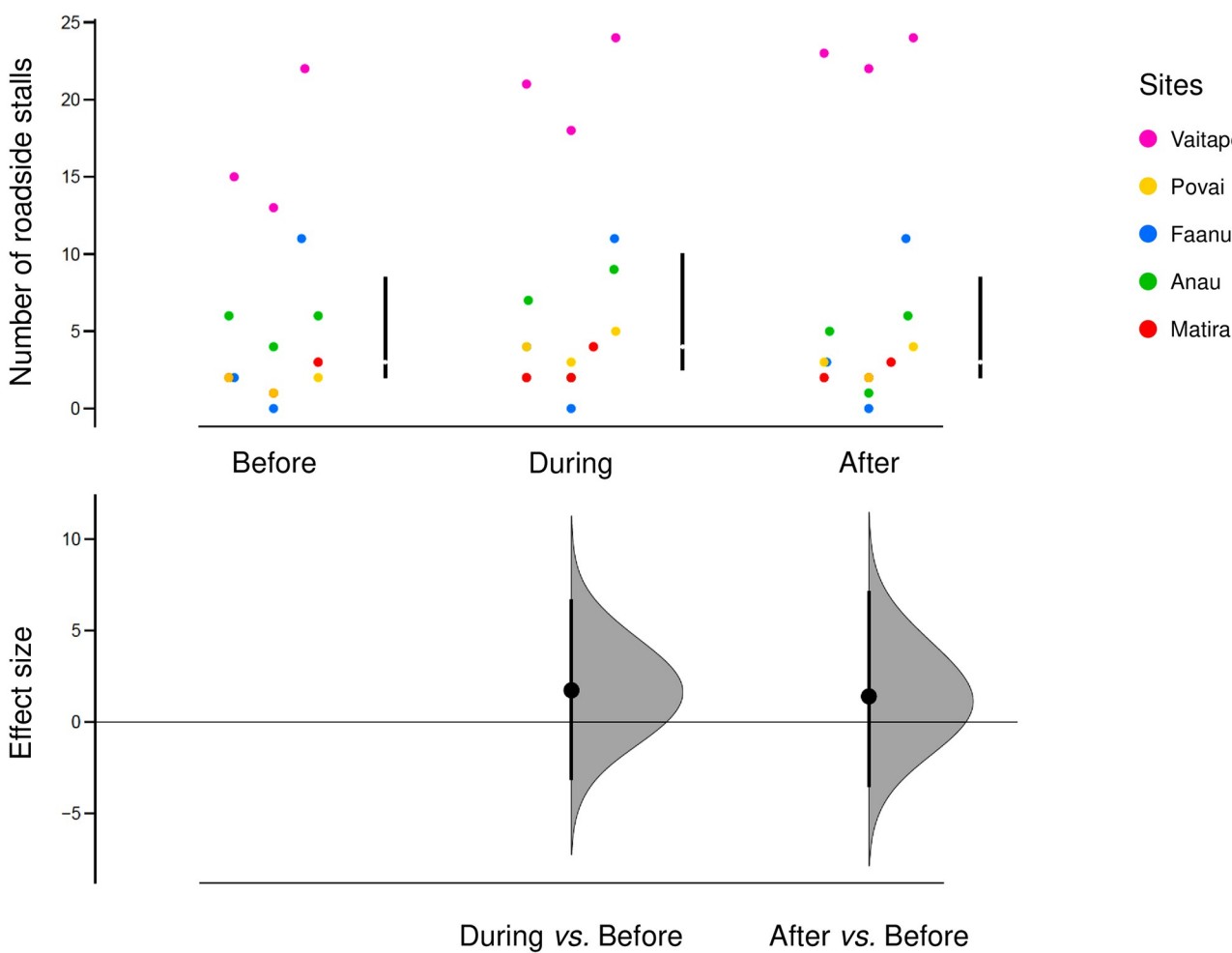

**Fig 2. Cumming estimation plot of the number of roadside stalls in Bora-Bora.** Top: Raw data per site with vertical bars representing the median (gap in the line) and 25th and 75th percentiles of each period. Bottom: Effect sizes measured as the paired mean differences (circle) ± 95% CIs (vertical bars). Sampling distributions are displayed in grey.

95% CI = -3.8–9.98; after: N = 5, mean difference = 1.8, 95% CI = -5–11.6). Similarly, the number of roadside stands selling fish from Monday to Sunday was slightly higher during the pandemic (Table 1c), but did not vary significantly (during: N = 5, mean difference = 1, 95% CI = -9.2–11.4; after: N = 5, mean difference = 0.4, 95% CI = -9.8–10.4) at the scale of Bora-Bora Island.

At the scale of districts, most stalls were located in Vaitape. There, the total number of all stalls increased during the pandemic (N = 3, mean difference = 4.33, 95% CI = -2–8.67) and was significantly higher after the pandemic than before it (N = 3, mean difference = 6.33, 95% CI = 0.67–9.67) (Fig 3). In Povai, the number of all stalls significantly increased also during the pandemic (N = 3, mean difference = 2.33, 95% CI = 1–3) and decreased after it, but there were still more stalls than before (N = 3, mean difference = 1.33, 95% CI = 0–2) (Fig 3). No differences were found in the other districts of Faanui (during: N = 3, mean difference = 0.67, 95% CI = -8–6.67; after: N = 3, mean difference = 0.33, 95% CI = -8–7), Anau (during: N = 3, mean difference = 0.67, 95% CI = -4–3; after: N = 3, mean difference = -1.33, 95% CI = -5–0.67) and Matira (during: N = 3, mean difference = 0.67, 95% CI = -0.67–2; after: N = 3, mean difference = 0.33, 95% CI = -1–1) (Fig 3).

**Table 1. Number of roadside stalls as a function of food products sold in each district of Bora-Bora and its evolution.**

*a. Roadside stands of land products from Monday to Saturday*

| Period | Vaitape | Povai | Faanui | Anau | Matira | Total |
|---|---|---|---|---|---|---|
| *Before COVID* | 13 | 2 | 2 | 4 | 1 | **22** |
| *During COVID* | 18 | 4 | 4 | 2 | 2 | **30** |
| *After COVID* | 22 | 3 | 3 | 1 | 3 | **32** |

*b. Roadside stands of cooked meals on Sunday*

| Period | Vaitape | Povai | Faanui | Anau | Matira | Total |
|---|---|---|---|---|---|---|
| *Before COVID* | 15 | 2 | 11 | 6 | 3 | **37** |
| *During COVID* | 21 | 5 | 11 | 9 | 4 | **50** |
| *After COVID* | 24 | 4 | 11 | 5 | 2 | **46** |

*c. Roadside stands of fish from Monday to Sunday*

| Period | Vaitape | Povai | Faanui | Anau | Matira | Total |
|---|---|---|---|---|---|---|
| *Before COVID* | 22 | 1 | 0 | 6 | 2 | **31** |
| *During COVID* | 24 | 3 | 0 | 7 | 2 | **36** |
| *After COVID* | 23 | 2 | 0 | 6 | 2 | **33** |

Before COVID: January and February 2020, During COVID: from March 2020 to October 2021, and After COVID: from November to December 2021.

## Discussion

The censuses conducted on Bora-Bora from January 2020 to December 2021 characterised the dynamic in the roadside sale of local products during the different COVID-19 pandemic periods (Table 1). This direct market system proved to be an essential source of income for the local population [20], especially in the context of reduced exports and border restrictions. Many people in touristic islands such as Bora-Bora have jobs related to the tourism industry and had to find an alternative source of income during the pandemic since unemployment rate drastically increased in 2020 as it has been reported by OECD [25]. During the lockdown, travel and movements were limited, and police restricted nautical activities, however the sale of roadside products in front of the seller's house was tolerated. Due to this economic

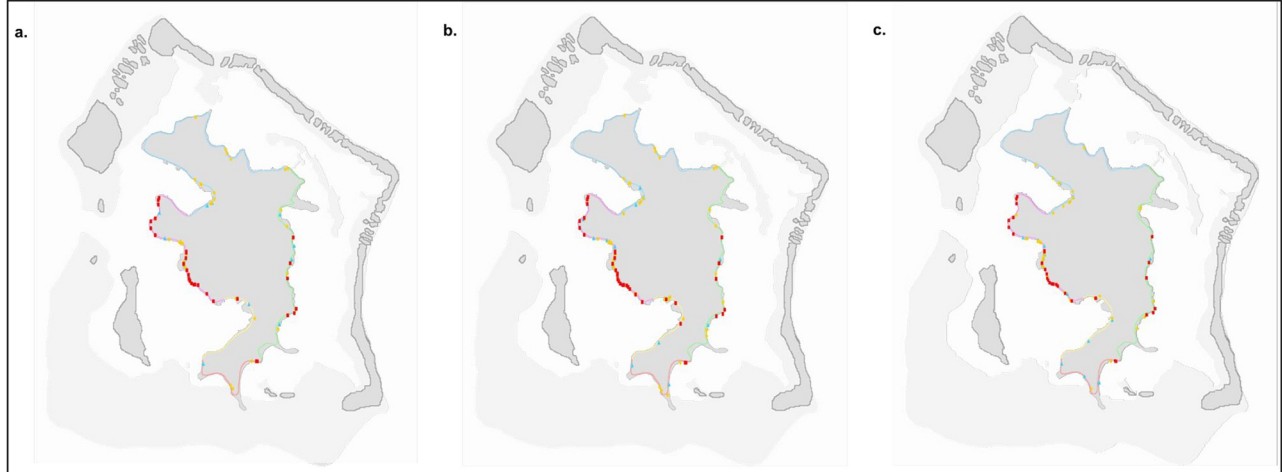

**Fig 3. Map of Bora-Bora showing the locations of the roadside stalls before (a), during (b) and after (c) the pandemic.** Stands of land products are represented by blue triangles, stands of cooked meals are represented by yellow circles and stands of fish are represented by red squares. Reprinted from an aerial photograph under a CC BY license, with permission from CRIOBE, original copyright 2018.

opportunity, the number of roadside vendors increased in the two Bora-Bora districts having the most of inhabitants (Vaitape and Povai). No changes were observed at the level of the overall island or at the level of the type of products that were sold. These results suggest that those districts, Vaitape and Povai, were either more in need than others or that people had to change their activity (from workers in hotels to fishermen and road sellers).

After November 2021, after the last lockdown of French Polynesia, economic activity resumed in Bora-Bora. Tourists returned to the hotels and tourism activities restarted. Nevertheless, the number of roadside sellers did not decrease in line with the increase in tourism activity (Table 1—data from French Polynesia Tourism Department: https://tahititourisme.fr/). Although Bora-Bora's economy picked up again after a long period of slowdown, there has been a shift in the economic activity of the local population. On the basis of informal interviews, some roadside sellers affirmed that they have moved from working in hotels to earn money from roadside stalls, with less pressure from any hierarchy. They are now self-employed or, sometimes employed by family members. For some, this is a supplement to their income and for others, it is a real change of profession, but we have not estimated the proportion of this supplementary income to the main income. The number of stalls can be interpreted as a proxy of changing economic activities, even if stalls are only one example of economic activities at Bora-Bora. The COVID-19 pandemic may have been an opportunity for workers to change their sector of activity in Bora-Bora and worldwide [16]. Lastly, the personal or professional reasons for this new activity were not always related to COVID-19. Some people declared that they had planned to set up a roadside stall beforehand and the pandemic only accelerated the process.

In conclusion, the COVID-19 pandemic highlighted the resilience of the food system in Bora- Bora and showed that diversifying food trade system is crucial [26], especially in isolated communities in SIDS and Pacific countries [27, 28]. Some informal discussions with sellers revealed that the pandemic has made people more aware of the precarity of their economic system, which is based solely on international tourism. Our study is essentially descriptive though and the results are supported only by some qualitative data, i.e., the statements we collected from vendors. However, this is related to the informal aspect of the food system we studied, which can sometimes be difficult to grasp at first glance. Nevertheless, it allowed us to highlight a modification, mainly as an increase in roadside sales activity due to the COVID-19 pandemic in two districts of Bora-Bora (Fig 2). Other informal food systems are also in place at Bora-Bora, as in many other SIDS and Pacific countries, such as subsistence fishing and cropping, gifted food, and bartering, but we have no quantitative data on it. However, our finding is in accordance with Fergusson et al. [29] which showed that local food production practices and food sharing conferred resilience in many Pacific Island, and that imported foods could aid or inhibit resilience. In the future, it would be interesting to complement this finding with quantitative/economic data and inferential statistics in order to, for example, analyse the evolution of sales prices and competition following the appearance of new stands. In addition, the amount of product sold by each vendor, including fish sales, could be examined to highlight the influence of the COVID-19 pandemic, or any other future event, acting on this food system, or on fishing pressures on the local reef ecosystem [30–32]. The financial factor should also be considered in particular by means of more formal interviews based on direct or semi-direct questions as many sellers claimed that their new activity was profitable.

## Supporting information

**S1 Table. Localisation and category of each roadside stall present in Bora-Bora during the three restriction periods: Before, during, and after the pandemic-related restrictions.**
(XLSX)

## Acknowledgments

We would like to thank the staff of 'Polynésienne des Eaux', 'Ia Vai Ma Noa Bora-Bora, the 'Commune de Bora-Bora' and CRIOBE (R. Madi Moussa, R. Galzin, T. Bambridge) for their help. All interviews followed the CNIL recommendations in France and the bioethical and environmental codes in French Polynesia (respect of anonymity and agreement of the interviewee for the information to be used for a research publication). All listed authors agreed on the publication of the present research and accepted responsibility for the work presented here. The authors have no competing interests to declare that are relevant to the content of this article. All data underlying the findings reported in the present study is already provided as a supplementary information file (S1 Table).

## Author Contributions

**Conceptualization:** David Lecchini.

**Formal analysis:** Alannah Gourlaouen.

**Funding acquisition:** Lana Minier, Vincent Sturny, David Lecchini.

**Investigation:** Lana Minier, Manon Fourrière, Emma Gairin, Alannah Gourlaouen, Cécile Berthe, Tehani Maueau, Milton Doom, Vincent Sturny.

**Resources:** Tehani Maueau, Milton Doom, Vincent Sturny.

**Supervision:** David Lecchini, Frédéric Bertucci.

**Visualization:** Manon Fourrière, Alannah Gourlaouen.

**Writing – original draft:** Alannah Gourlaouen, David Lecchini.

**Writing – review & editing:** Lana Minier, Manon Fourrière, Emma Gairin, Alannah Gourlaouen, Stéphanie Krimou, Suzanne C. Mills, David Lecchini, Frédéric Bertucci.

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
