## [Decision Letter · Decision Letter 0]

6 Sep 2022

PONE-D-22-15433Roadside sales activities in a South Pacific Island (Bora-Bora) reveal sustainable strategies for local food supply during a pandemicPLOS ONE

Dear Dr. Bertucci,

Thank you for submitting your manuscript to PLOS ONE. After careful consideration, we feel that it has merit but does not fully meet PLOS ONE’s publication criteria as it currently stands. Therefore, we invite you to submit a revised version of the manuscript that addresses the points raised during the review process.

Two reviews have been received of your manuscript with both reviewers highly experienced socio-economic scientists working in developing countries. They both concur that a major revision is required before the manuscript would be re-considered for publication. So please address the concerns raised by both reviewers with particular emphasis on the many points raised by Reviewer 2. I look forward to seeing a revised and much improved version for re-submission.

We look forward to receiving your revised manuscript.

Kind regards,

Andrew Halford

Academic Editor

PLOS ONE

Journal Requirements:

3. We note that Figure 1 your submission contains [map/satellite] images which may be copyrighted. All PLOS content is published under the Creative Commons Attribution License (CC BY 4.0), which means that the manuscript, images, and Supporting Information files will be freely available online, and any third party is permitted to access, download, copy, distribute, and use these materials in any way, even commercially, with proper attribution. For these reasons, we cannot publish previously copyrighted maps or satellite images created using proprietary data, such as Google software (Google Maps, Street View, and Earth). For more information, see our copyright guidelines: http://journals.plos.org/plosone/s/licenses-and-copyright.

Reviewers' comments:

Reviewer's Responses to Questions

**Comments to the Author**

1. Is the manuscript technically sound, and do the data support the conclusions?

Reviewer #1: Partly

Reviewer #2: Partly

2. Has the statistical analysis been performed appropriately and rigorously? 

Reviewer #1: I Don't Know

Reviewer #2: N/A

3. Have the authors made all data underlying the findings in their manuscript fully available?

Reviewer #1: Yes

Reviewer #2: No

4. Is the manuscript presented in an intelligible fashion and written in standard English?

Reviewer #1: Yes

Reviewer #2: Yes

5. Review Comments to the Author

Reviewer #1: Dear authors,

Thank you for the opportunity to review your manuscript. Please see below for my comments and questions with regards to your submission.

Ethics? The authors refer to conducting short semi-directed interviews. Could they provide a brief statement in the materials/methods as to why ethics was not required. The results provided in the manuscript do not rely on the interviews but comments in the discussion highlight that interviews gathered information from stall owners, information being used for a research publication.

Line 51: a note could potentially be added to explain French Polynesia is a Pacific Island Territory, for people unfamiliar with the region

Lines 51-53: add details about the different restrictions as it was unclear as to whether total lockdown included also a ban of foreign tourists…this information is important in the light of the discussion where tourism is brought up quite extensively. Did the lockdowns include an international and domestic travel ban. Could the authors specify if the ban of foreign tourists was inclusive of a ban of French nationals, or mention ban on non-French Polynesian residents…

Line 64: Is any official data on unemployment, part-time employment available to complement the statement?

Line 90: Could the authors be more specific as to what they mean by “several” censuses.

Materials and methods

Could the authors provide more information on dates or frequency of census undertaken before/during/after the covid-19 social and travel restrictions. Out of the “at least 65 round trips”, how many surveys were conducted before/during/after during weekdays and how many (for a total of 16) were conducted before/during/after on Sundays. I understand that the authors aimed to report the maximum number of roadside stalls for each category per period but the details are currently insufficient to allow other researchers to replicate the study.

Could the authors provide details as to the reasons why a GPS coordinate was the criterion for recording a new vs an absent stand? Could families travel in different locations in the same districts on different days, at different times (although restricted during Covid-19) or were families only putting stalls in front of their houses? How is the information known by the authors? The authors do not mention that names or addresses were taken from people manning stalls.

I think it would benefit the manuscript to understand how the authors accounted for the fact that their different time periods were of different lengths (apart from the before/after pandemic which were roughly similar in length). I understand the authors compared their method for reporting the maximum number of stalls instead of average number to population census. However, census of population tend to report a count of people on a given day not over a period as in this study. Could this be more clearly explained?

Results

I would suggest the authors include the percentage difference in brackets in the different tables to make references and comparisons easier.

Discussion

I believe the authors could add elements to their discussion which seems to highlight people choosing a livelihood diversification strategy. How does it compare with other relevant literature from before or during the pandemic? What further research would the authors suggest to follow up from their study?

The point the authors make on stalls selling fishery product could be discussed further in the context of other studies around the region which looked at the impact of Covid-19 on fishing for example. The authors could use their results to provide recommendations to fishery management agencies as part of their ongoing monitoring system, in light of the importance of fishery resources to Pacific people.

General comment: There are some minor English language errors in the text.

Reviewer #2: Comments are included in attached document. I copy the first part of the document here just because the system forces me to write here.

Economic data tend to be based on official numbers, but informal systems can be as important, particularly in developing countries. It is important to understand informal food production and trade systems as it helps understand strengths and limitations to improve food security and resilience in the face of different disasters, including covid, but also climate change related impacts. This article contributes to building such understanding.

As the authors mention, their quantitative data is limited, but these data, added to the potential of qualitative data from interviews, can provide an interesting picture of the informal trade system of Bora-Bora island, particularly their roadside stalls. Methods need to be improved to allow readers to understand what was done and why.

Language in general is good, allowing the reader to understand what the authors intend to explain. Some sections can be improved, but unfortunately doing so on a PDF is too inefficient, so I limit my comments to some unclear sections. It would be good however, to get an English edition just before re-submission.

Introduction: Could you please provide some more context about the study? Was this part of a larger study or a stand-alone study? If it is complementary to other research you have been doing, it would be interesting to make the connections through a short description or references. I assume road stalls are only one of many options of trading/accessing food. What other informal food systems are in place (a short reference to for example importance of subsistence fishing/cropping, gifted food, bartering?). Such connections might also be relevant to include in the discussion section.

Methods: This section needs to be improved to allow readers to replicate the study if they wanted to. In particular, you mention interviews, but these are treated almost as an afterthought – it’s impossible for the reader to find out what was said by interviewees and what are the authors’ insights or opinions. Nevertheless, if done properly, interviews are an important source of information. Enough detail needs to be provided in this section about them: did you conduct interviews during all your surveys (survey meaning a sampling effort – e.g. roundtrip)? how did you know which stall holders you were going to interview or not during each survey? Did you have some criteria to decide if you were interviewing that person or not (e.g. older than 18?); how did you ensure you covered the diversity of views you wanted to cover (e.g. different types of stalls, provinces, gender, ages)? How long did the interviews last? Did you interview people more than once to understand changes? Provide a link to the interview guides, so that the reader understands how you approached your themes (did you have themes?) or just the general topic. Were interviews conducted in French? Translated? Transcribed? How did you analyse the interviews? Did you code the interviews? Deductively or inductively? If this is the first time you use qualitative data, you can get an idea of what qualitative analysis is here https://gradcoach.com/qualitative-data-analysis-methods/. Include also any key ethics considerations, like informed consent or if you interviewed minors, did you request parents’ authorisation?

From what I can gather, you decided to use the results from your interviews to enrich the discussion, in addition to references… Please explain why you didn’t present the interview data in the results section. In the discussion the source of information becomes unclear, as you are mixing up your views, interview results and references. So I would rather have a section within results where you present interview data. But it’s your decision, depending on your data.

Results: How many people did you interview? (from each type of stall, from each province? From each gender? Age group?). You discuss that stall holders moved towards Vaitape and Povai, and you also have GPS coordinates – can you show the location of stalls (and changes?) on your map(s)?. Interview results would be enriched if you provided a few quotes to illustrate key points. In the section where you present number of stalls, proportions and changes, try to be consistent, so that it’s easier for the reader to follow your key points, rather than having to move up and down to understand if x proportion is equivalent to the proportion mentioned before for another type of stall.

Discussion – The use of sources needs to be revised. For example, some statements are not properly referenced: e.g. “Many people in Bora-Bora have jobs related to the tourism industry and had to find an alternative source of income during the pandemic [11]”. This reference, however, is not about Bora-Bora and the main points are that the loss of tourism in dependent countries has resulted in increased poaching and extractive industries taking advantage… If there are no references in Bora-Bora of whatever idea you are proposing, but you find that it happened elsewhere, you would need to say something like “…, as has been reported elsewhere/name of place.”. Review all references to make sure that the statements correspond to the reference, and if not, clarify the relationship between the statement and the reference provided. If interview results are kept in this section, you will also need to clarify which statements are from this study (according to one/some/most of our interviewees… or provide a short quote and in brackets a code for that interviewee; e.g. female “tui” stall holder from Vaitape. If on the other hand you are proposing an idea, you can clarify to the reader that it is yours “We propose here that x happened because of y” (I am unsure of the journal politics regarding passive voice vs. direct statements, so re-write accordingly if necessary).

Study limitations – as I am not 100% sure about your sampling strategy, I don’t know why you didn’t record certain data on the stall when you interviewed people (e.g. amount of fish, crops, people manning the stall). If not included already, mention all key items to record in potential future studies, and explain why you didn’t recorded them (can be in methods if more appropriate).

Availability of data: the authors state that the data are fully available upon request, but this hasn’t been stated in the paper.

See attached document for specific comments.

6. PLOS authors have the option to publish the peer review history of their article (what does this mean?). If published, this will include your full peer review and any attached files.

Reviewer #1: No

Reviewer #2: **Yes: **Carolina Garcia

---

## [Author Response · Author response to Decision Letter 0]

15 Feb 2023

Comments of reviewer 1:

1/ Ethics? The authors refer to conducting short semi-directed interviews. Could they provide a brief statement in the materials/methods as to why ethics was not required. The results provided in the manuscript do not rely on the interviews but comments in the discussion highlight that interviews gathered information from stall owners, information being used for a research publication.

We are sorry to have misled the two reviewers about these interviews. Indeed, we conducted some interviews. However, these interviews were more informal discussions with some road sellers during our different surveys, than real interviews to understand the fishery in Bora. Therefore, we deleted the information about interviews in the Methods section, and we added some precisions about these informal discussions in the Discussion. In the acknowledgments part, we added “All interviews followed the CNIL recommendations in France and the bioethical and environmental codes in French Polynesia (respect of anonymity and agreement of the interviewee for the information to be used for a research publication).”

2/ Line 51: a note could potentially be added to explain French Polynesia is a Pacific Island Territory, for people unfamiliar with the region

We added this information in the revised ms (Line 60).

3/ Lines 51-53: add details about the different restrictions as it was unclear as to whether total lockdown included also a ban of foreign tourists…this information is important in the light of the discussion where tourism is brought up quite extensively. Did the lockdowns include an international and domestic travel ban. Could the authors specify if the ban of foreign tourists was inclusive of a ban of French nationals, or mention ban on non-French Polynesian residents…

We added more explanation in the revised ms (Lines 71-76).

4/ Line 64: Is any official data on unemployment, part-time employment available to complement the statement?

Unfortunately, there is not official report about the COVID effects on tourism in French Polynesia. The present data came from the French Polynesia Tourism Department: https://tahititourisme.fr/ (Lines 84 and 203).

5/ Line 90: Could the authors be more specific as to what they mean by “several” censuses.

We removed this term.

6/ Materials and methods. Could the authors provide more information on dates or frequency of census undertaken before/during/after the covid-19 social and travel restrictions. Out of the “at least 65 round trips”, how many surveys were conducted before/during/after during weekdays and how many (for a total of 16) were conducted before/during/after on Sundays. I understand that the authors aimed to report the maximum number of roadside stalls for each category per period but the details are currently insufficient to allow other researchers to replicate the study.

We rephrased this as “Stalls were surveyed 9 times during the week (Monday to Saturday) and 3 times on Sunday before COVID-19 period; 16 times during the week and 7 times on Sunday during COVID-19 pandemic; and 11 times during the week and 4 times on Sunday after COVID-19 period.” (Lines 118-121). 

7/ Could the authors provide details as to the reasons why a GPS coordinate was the criterion for recording a new vs an absent stand? Could families travel in different locations in the same districts on different days, at different times (although restricted during Covid-19) or were families only putting stalls in front of their houses? How is the information known by the authors? The authors do not mention that names or addresses were taken from people manning stalls.

The road sellers always put stalls in front of their houses or their boat. If they had to move from their house/boat to another site, although it was rare (except due to COVID pandemic as shown in our study), they set up their new stand in front of the house of one member of their family. Therefore, GPS data is an excellent proxy of the road stand numbers. Thus, to identify each roadside stall and not count the same stall twice, we asked for the name of the person (as the island is small, we often knew the person as well), we recorded the GPS coordinates, or we described the color of the fisherman boat. We would also ask the seller if their stand was always in the same location.

8/ I think it would benefit the manuscript to understand how the authors accounted for the fact that their different time periods were of different lengths (apart from the before/after pandemic which were roughly similar in length). I understand the authors compared their method for reporting the maximum number of stalls instead of average number to population census. However, census of population tend to report a count of people on a given day not over a period as in this study. Could this be more clearly explained?

From our perspective, the number of stalls we report is independent of the length of the different periods. Determining the number of people on a given day when looking at changes in a cumulative number is one way to account for the possibility that the same person will be counted several times during the sample period. The possibility of this happening over a long period of time is indeed more important and must be corrected for duration. Here, stalls can only be counted once over the entire sample period and we therefore report an absolute total number of stalls present at any time in Bora Bora. We now specify that we report an absolute number of stalls present at any time on the island, which is independent of the duration of the sampling periods (Lines 109 and 121).

9/ Results. I would suggest the authors include the percentage difference in brackets in the different tables to make references and comparisons easier.

We considered a new data analysis and now compare the differences between periods by calculating the effect sizes of statistical measures. To answer this comment, we would have to make a second table because the % represent either the number of stalls of a village compared to all those on Bora-Bora, or a difference in the number of stalls between the three COVID periods. 

10/ Discussion. I believe the authors could add elements to their discussion which seems to highlight people choosing a livelihood diversification strategy. How does it compare with other relevant literature from before or during the pandemic? What further research would the authors suggest to follow up from their study? The point the authors make on stalls selling fishery product could be discussed further in the context of other studies around the region which looked at the impact of Covid-19 on fishing for example. The authors could use their results to provide recommendations to fishery management agencies as part of their ongoing monitoring system, in light of the importance of fishery resources to Pacific people.

We modified the introduction to take into consideration this comment and cite different papers/studies focusing on how the social restrictions impacted the implementation of Sustainable Development Goals related to income, nutrition, and food security. In our conclusion, we underlined the importance of the diversifying food trade system especially in isolated communities in SIDS and Pacific countries, and also how the present study could be improved if another world social crisis happens. Also, these data on roadside sales were used in a parallel study on the characterization of lagoon fishing (self-consumption of fish by Polynesians vs. selling in hotels). The amateur fishermen sell on the roadside only, only professionals used to sell in hotels but this had changed during the COVID-19 crisis (100% of the sales were made on the roadside because hotels were closed).

Comments of reviewer 2:

1/ Economic data tend to be based on official numbers, but informal systems can be as important, particularly in developing countries. It is important to understand informal food production and trade systems as it helps understand strengths and limitations to improve food security and resilience in the face of different disasters, including covid, but also climate change related impacts. This article contributes to building such understanding. As the authors mention, their quantitative data is limited, but these data, added to the potential of qualitative data from interviews, can provide an interesting picture of the informal trade system of Bora-Bora island, particularly their roadside stalls. Methods need to be improved to allow readers to understand what was done and why. Language in general is good, allowing the reader to understand what the authors intend to explain. Some sections can be improved, but unfortunately doing so on a PDF is too inefficient, so I limit my comments to some unclear sections. It would be good however, to get an English edition just before re-submission. 

Thanks for this general positive comment.

2/ Introduction: Could you please provide some more context about the study? Was this part of a larger study or a stand-alone study? If it is complementary to other research you have been doing, it would be interesting to make the connections through a short description or references. I assume road stalls are only one of many options of trading/accessing food. What other informal food systems are in place (a short reference to for example importance of subsistence fishing/cropping, gifted food, bartering?). Such connections might also be relevant to include in the discussion section.

We specified in the introduction that “The present study was opportunistic…” as it is not related to other research (Line 85).

Subsistence fishing/cropping, gifted food, or bartering are also present at Bora-Bora, but we have no scientific and quantitative data to present in the present paper. Nevertheless, we added this information in the Discussion part of the revised manuscript.

3/ Methods: This section needs to be improved to allow readers to replicate the study if they wanted to. In particular, you mention interviews, but these are treated almost as an afterthought – it’s impossible for the reader to find out what was said by interviewees and what are the authors’ insights or opinions. Nevertheless, if done properly, interviews are an important source of information. Enough detail needs to be provided in this section about them: did you conduct interviews during all your surveys (survey meaning a sampling effort – e.g. roundtrip)? how did you know which stall holders you were going to interview or not during each survey? Did you have some criteria to decide if you were interviewing that person or not (e.g. older than 18?); how did you ensure you covered the diversity of views you wanted to cover (e.g. different types of stalls, provinces, gender, ages)? How long did the interviews last? Did you interview people more than once to understand changes? Provide a link to the interview guides, so that the reader understands how you approached your themes (did you have themes?) or just the general topic. Were interviews conducted in French? Translated? Transcribed? How did you analyse the interviews? Did you code the interviews? Deductively or inductively? If this is the first time you use qualitative data, you can get an idea of what qualitative analysis is here https://gradcoach.com/qualitative-data-analysis-methods/. Include also any key ethics considerations, like informed consent or if you interviewed minors, did you request parents’ authorisation? 

We are sorry to have misled the two reviewers about the interviews. Indeed, we conducted some interviews. However, these interviews were more informal discussions with some road sellers during our different surveys, than real interviews to understand the fishery in Bora. Therefore, we deleted the information about interviews in the Methods section, and we added some precisions about these informal discussions in the Discussion (Lines 205-215).

4/ From what I can gather, you decided to use the results from your interviews to enrich the discussion, in addition to references… Please explain why you didn’t present the interview data in the results section. In the discussion the source of information becomes unclear, as you are mixing up your views, interview results and references. So I would rather have a section within results where you present interview data. But it’s your decision, depending on your data.

See previous comment.

5/ Results: How many people did you interview? (from each type of stall, from each province? From each gender? Age group?). You discuss that stall holders moved towards Vaitape and Povai, and you also have GPS coordinates – can you show the location of stalls (and changes?) on your map(s)?. Interview results would be enriched if you provided a few quotes to illustrate key points. In the section where you present number of stalls, proportions and changes, try to be consistent, so that it’s easier for the reader to follow your key points, rather than having to move up and down to understand if x proportion is equivalent to the proportion mentioned before for another type of stall. 

See previous comment about interviews.

In the discussion part, we modified our interpretation of the data between the villages. Indeed, stall holders could move towards Vaitape and Povai, but it could be also some new stall holders in Vaitape as some local inhabitant lost their job in the hotels.

6/ Discussion – The use of sources needs to be revised. For example, some statements are not properly referenced: e.g. “Many people in Bora-Bora have jobs related to the tourism industry and had to find an alternative source of income during the pandemic [11]”. This reference, however, is not about Bora-Bora and the main points are that the loss of tourism in dependent countries has resulted in increased poaching and extractive industries taking advantage… If there are no references in Bora-Bora of whatever idea you are proposing, but you find that it happened elsewhere, you would need to say something like “…, as has been reported elsewhere/name of place.”. Review all references to make sure that the statements correspond to the reference, and if not, clarify the relationship between the statement and the reference provided. If interview results are kept in this section, you will also need to clarify which statements are from this study (according to one/some/most of our interviewees… or provide a short quote and in brackets a code for that interviewee; e.g. female “tui” stall holder from Vaitape. If on the other hand you are proposing an idea, you can clarify to the reader that it is yours “We propose here that x happened because of y” (I am unsure of the journal politics regarding passive voice vs. direct statements, so re-write accordingly if necessary).

We followed reviewer’s suggestion (Lines 190-193) and now refer to the work of OECD. Such information for Bora Bora was actually reported during the pandemic from the French Polynesia Tourism Department: https://tahititourisme.fr/. There were several articles in the two Polynesia newspaper (Tahiti Infos, La Dépêche) and also several reports from French Government about the economic impact of COVID in French Polynesia, but unfortunately, there is no scientific publications.

7/ Study limitations – as I am not 100% sure about your sampling strategy, I don’t know why you didn’t record certain data on the stall when you interviewed people (e.g. amount of fish, crops, people manning the stall). If not included already, mention all key items to record in potential future studies, and explain why you didn’t recorded them (can be in methods if more appropriate). 

In the revised Methods section, we have clarified our surveys of the roadside stall types. Note that we now consider only three types of stall (land products, cooked meals and fish).

8/ Availability of data: the authors state that the data are fully available upon request, but this hasn’t been stated in the paper. 

We now added the information in the acknowledgments section.

Specific comments:

9. An Ethics statement is missing. As the study involved human participants (interviews), this is a requirement, even if the organisation/country doesn’t require one (which is rare nowadays). The authors need to check if their organisation and/or country do have a human ethics approval system that they were not aware of; if that is the case, this step would have to be fulfilled (include approval number/code if that’s the case); if not possible, explain what steps were taken to protect the rights of participants. The minimal information to include is that explanation of the purpose of the project, benefits/risks and treatment of personal information was provided, and that all participants agreed to participate (informed consent). Include a short statement in the methods section, or wherever the journal suggests.

In the acknowledgment part, we added “All interviews followed the CNIL recommendations in France and the bioethical and environmental codes in French Polynesia (respect of anonymity and agreement of the interviewee for the information to be used for a research publication).”

10/ Row 23-25: “In many Pacific Islands, people returned to nature in order to provide a much-needed source of food for the local population” – it is unlikely that this was the reason for Pacific Islanders to “return to nature” – most likely they did so to provide for themselves, their families, or to generate income.

We modified the sentence as suggested by the reviewer (Lines 22-23).

11/ Row 42: “Small Island Developing States (SIDS) are one region particularly vulnerable to global social, economic and environmental crises” – SIDS is not a region – there are SIDS in different regions (e.g. the Pacific, the Caribbean… - correct sentence)

We modified the sentence as suggested by the reviewer.

12/ Row 48: Were exports blocked in French Polynesia? The references seem to be from other places and is presented as an important argument. If people need to turn to local resources to meet food requirements – is it because exports are blocked, or imports? Maybe you can use “trade disruptions” instead, a more general term. But please include the direct link to “increasing pressures on local resources” – is it because imported food was less available? Or maybe the real reason is because more people had income shortages, and local food is likely to be cheaper than imported food (or almost free if you fish/produce it yourself)? I suggest to re-write this sentence including a link to the Polynesian context in particular… 

We follow this comment and have totally modified the last part of this paragraph (Lines 36-53).

13/ Row 93: Can you provide more context about the differences between the provinces? Some indications are scattered through the document, but some basic information should be provided early on, with the map, like population; economic status if available; main activities including tourism, fishing and agriculture; road quality/access…

Except this information “Vaitape is the main village on the island and the most touristic part of Bora-Bora along with Povai and Matira. Anau and Faanui are more rural parts of the island. There are no sales of land or sea products on the different islets surrounding the main island of Bora-Bora. The only road is the one that runs along the coast all around the island.”, there is no more information about these different villages and districts. There is no official report from Government or scientific publication to give more information.

14/ Row 95: The map provided seems to have some holes (coloured contours don’t match up) – are these landlocked provinces? Please include the road(s) that you followed in your map, and the location of the stalls (it might be necessary to show a map for each period?). Include an inset with a general location of Bora-Bora in relation to a more general world/region map. The quality of the map could be improved.

We produced a new figure which better shows the road along the coastline. We modified the legend of the figure. The new figure has been uploaded in better quality. We added the location of the stands on the map based on GPS coordinates at each Covid period on a new figure (Figure 3). Coordinates are provided in a Table as a supplementary material.

15/ Row 103: Semi-directive interviews? Do you mean semi-structured? The use of interviews is poorly explained both in methods and in results. If interviews were used, keep in mind that this is one of a collection of respected social-science methods (albeit mostly qualitative in nature): the design of the study and the reporting of the associated methods (sampling, implementation, analysis) needs to be included to legitimise the data collected. I suggest following a guiding document on use and report on interview results (a simple one in Young et al 2017 - A methodological guide to using and reporting on interviews in conservation science research; more in-depth: Sage publications on qualitative research in general, or interviews in particular): Key information to include (from Young et al). Here, the following information is missing: sampling strategy (did you interview all stall holders or did you try to cover a diversity of them (e.g. the different types of stalls described, different genders, different age categories, different districts? Different sampling periods?); how many people did you interview from each criteria (at least types of stalls); did you repeat interviews with the same participants for the different periods?; did you conduct a pilot to test the interview guide?; provide access to the interview guide as supplementary material or reference to where people can find it; methods in the field – consent before starting interview? interviews were carried out at the stalls? How long did they last? Did you record them and transcribe them? How did you analyse the information from interviews? Coding strategy? Other ethical considerations: how did you ensure that the data stays anonymous and confidential.

We are sorry to have misled the two reviewers about these interviews. Indeed, we conducted some interviews. However, these interviews were more informal discussions with some road sellers during our different surveys, than real interviews to understand the fishery in Bora. Therefore, we deleted the information about interviews in the Methods section, and we added some precisions about these informal discussions in the Discussion. In the acknowledgment part, we added “All interviews followed the CNIL recommendations in France and the bioethical and environmental codes in French Polynesia (respect of anonymity and agreement of the interviewee for the information to be used for a research publication).”

16/ Rows 105-118: the use of the term “Survey” is confusing here… This is more a classification of the type of stalls that you surveyed… So you probably mean groups, type of stalls, or something like that. You’ve already used survey to mean (I think, it needs clarifying as well) 1 round-trip around the island… Change the term throughout the document. 

Thanks for this comment. We deleted the term ‘Surveys’ in the revised ms.

17/ Rows 127-129: The sampling strategy needs to be clarified: it says that only new/absent stalls were identified… However, you must have had a baseline at an early point to decide that a stall was new or absent; this is not explained.

In the revised Methods section, we have clarified our surveys of the three roadside stall types.

18/ Row 130: You say “the three census periods”, but up to this point you haven’t explained what these are (other than in the abstract, which is independent). You can improve the description of all the different combinations (i.e. “period” x “survey” x “round-trip/district targeted trip” x “district” x “week/weekend day”) – A table could be easier to understand for your readers and would avoid a long text to try to put in all that info, but this is just a suggestion. I am assuming here that each time you went out, you recorded information on any kind of stall, and not that you had cooked meals days and ignored fish stalls those days? If that’s not the case, you would need to include that information as well…

In the revised Methods section, we have clarified our surveys of the three roadside stall types. “Our goal was to identify the maximum absolute number of roadside stalls as possible present at any time in Bora-Bora during the three restriction periods: before, during, and after the pandemic-related restrictions. As the presence of a same stall may vary during the day, on different days of the week, and between weeks for various reasons (inefficient fishing, rain limiting fruit harvesting, plant collection period, holidays, etc.), the censuses were conducted randomly at different times of the day, on different days and over different weeks, either all around the island or on a preferential district (depending on the number of stalls). The absence of a stall during a tour of the island did not mean it was considered absent. For each tour of the island, we noted all the stalls present, regardless of the food sold. Once back at the laboratory, we checked if the observed stalls had already been noted in our Excel database. Stalls were surveyed 9 times during the week (Monday to Saturday) and 3 times on Sunday before COVID-19 period; 16 times during the week and 7 times on Sunday during COVID-19 pandemic; and 11 times during the week and 4 times on Sunday after COVID-19 period.”. 

19/ Row 131: No need to repeat “similar to human population censuses” – this has been said and it’s not particularly important. Check for other instances in the rest of the paper for repeated/unnecessary information…

We follow this comment and have deleted this repetition.

20/ Rows 134-136: descriptive statistics can be thorough… did you mean inferential statistics? I feel you are trying to justify why your study didn’t include more difficult statistics. I believe that a descriptive study, supported by qualitative data (interviews) is valuable enough, particularly because the stalls are part of an informal food system – key for the local populations as income generators and as a contributor to food security; sometimes more important than “official” systems! Informal systems are always more difficult to track with traditional quantitative methods, and for this reason descriptive analysis and qualitative data become even more important. This doesn’t mean, however, that certain protocols can be ignored, to show to your readers that descriptive statistics and qualitative methods were designed and implemented to ensure your results are valid (see comment above about interviews). Limitations of a study are usually presented at the end of a paper, where you provide recommendations for future studies. 

Thank for this comment. We added some sentence about study limitations in the discussion part. However, we now consider a new data analysis and we compare the differences between periods by calculating the effect sizes of statistical measures (Lines 124-128).

21/ Row 139: …, on average 63%, most stalls… the 63% should be in brackets; however, I’d consider getting rid of this average here, saying simply that most stalls were in Vaitape… For me, the evolution of this proportion (59 to 60 to 69%) is more interesting and shows me at the same time that the proportion was high for each period and overall… Consider doing the same for the other stall types.

We followed this comment to modify the results section.

22/ Row 160: The table provides information about the 4 types of stalls – put it either at the beginning of the descriptions (my preference) or at the end, rather than in the middle… If you put the table at the beginning, include a short reference to it in the text just before.

To follow this comment, we moved the table in the Results section.

23/ Rows 169- 171: add “respectively” to clarify that 44% and 50% refer to Anau and Matira, and clarify that it refers to the change between period 2 and 3.

This sentence no longer appears in the revised version of the manuscript.

24/ Row 177: when you talk about ‘tui’ sellers, do you mean only people who sell fish bundles, or any fish seller? As you said before that ‘tui’ is “bundle”, you would need to avoid the term tui seller, or clarify in the methods that you will be referring to both types of fish sellers (bundle AND bags) as “tui sellers” (if that’s how everybody calls them, regardless of how they actually sell the fish?).

To avoid misunderstanding, we deleted the term ‘tui’ sellers.

25/ Rows 183-185: Saying that the changes were mainly in Povai is a bit misleading, as they started with only 1 stall, meaning that ANY change will be significant (1 more and that’s already 100%). Check if for the rest of the text, there are other misleading conclusions, where it might be better to talk about numbers rather than proportions… 

We followed this recommendation to avoid such a misunderstanding and rather base our conclusions on the effect sizes we measured.

26/ Row 190: It’s not clear if the 55% increase refers to the overall increase (period 1 to period 3)? Try to follow the same pattern when you present the results for each stall type, so that it’s easier to the reader to understand the main results.

We substantially revised the results section and now present the data in a more consistent way.

27/ Row 191-193: Try not to repeat the information in the table and in the text – most people won’t be familiar with the provinces’ names, so it’s quite hard to follow what you are trying to say (i.e. 2 new here, one here and another there, then one less over there…) – this information is instead quite clear in the table.

As suggested by the reviewer, we only refer the reader to the table and substantially revised the entire section.

28/ Row 196: It’s not the distribution that changed (as long as we know!), but the number of roadside stalls. 

We modified the manuscript to avoid such a mistake.

29/ Row 206: Did you mean direct rather than indirect? Sellers are mostly selling directly to end-consumers, aren’t they?

Thanks for this comment. We use the term ‘direct’ (Line 189).

30/ Row 213-215: What is the source of this information? Interviews? Reference?

We deleted these sentences in the revised ms.

31/ Row 217: quantity or frequency of fish … not sure what frequency refers to? The number of times they (stall holders) were on the road each week for example? Clarify, as putting it together with quantity can be confusing, as the measurements are likely to be quite different…

We deleted the sentence and restructure the section to avoid such a confusion.

32/ Row 226-227: Maybe from your interviews, do you have more information about the sellers for whom the stalls represent supplementary income vs. main income?

Sorry, we have not this information (supplementary income vs. main income).

33/ Row 227-229: Check what you meant by ‘…towards in Vaitape…’. In addition to checking this sentence, the results presented in this paper don’t include information on shifting economic activities – are you referring to data from interviews? In that case, you need to present that. Or are you proposing that the number of stalls can be interpreted as changing economic activities (stalls are only one example of “economic activities”, but you think it can be a proxy?)? If so, explain your arguments.

Row 231: “may also be a reason” – it’s common around the world to find more sales where there’s more people, so you can get rid of the doubt in this sentence…

Row 232: What is the significance for the rest of the paragraph or the study that Matira is famous for its beaches? Explain

Row 233-235: What does “moved” mean? Do they move their stalls (and they go back and forth each day they have their stall open), or they migrated permanently (with their families?)? How do you know that products are mostly for locals (during border closures that seems obvious, but after?)? What is the source of all this information? Interviews?

We followed these 4 comments to modify the sentences about the shifting economic activities in the revised ms.

34/ Row 235: either …pandemic may have been as an opportunity…, or …pandemic may have been SEEN as an opportunity…

We modified the sentence as “The COVID-19 pandemic may have been an opportunity for workers to change their sector of activity in Bora-Bora and worldwide” (Lines 211-212).

35/ Row 236: I’d start a new paragraph here, from “tourism activity…”, to discuss what might happen in the future and discuss in more detail the importance of considering the resilience of the population in the face of future events (including climate-change related) – the response of the local population to a scarcity of jobs seems like a positive trait that would help them prepare for other major events? Check arguments presented in Ferguson et al 2022, Marine Policy 137 - Local practices and production confer resilience to rural Pacific food systems during the COVID-19 pandemic.

Row 243: While not directly related to your study because you don’t seem to have data to support that marine habitats and species were under higher pressure, studies outside Bora-Bora, around the Pacific have shown that it was the case for many places (e.g. Fergusson et al 2022). You can find further references that support diversifying food production and income activities and elaborate on this point as a separate paragraph.

We modified the introduction part to cite different papers/studies how the social restrictions impacted the implementation of Sustainable Development Goals related to income, nutrition, and food security. In our conclusion, we underlined the importance of diversifying food trade system especially in isolated communities, in SIDS and Pacific countries, and also how the present study could be improved if another crisis happens. Lastly, we added the paper of Fergusson et al. in this section.

36/ Row 245-248: These two sentences would be better placed with a previous paragraph that talked about the shift of economic activities. 

We moved these two sentences.

---

## [Editor Report · Decision Letter 1]

28 Mar 2023

Roadside sales activities in a South Pacific Island (Bora-Bora) reveal sustainable strategies for local food supply during a pandemic

PONE-D-22-15433R1

Dear Dr. Bertucci,

We’re pleased to inform you that your manuscript has been judged scientifically suitable for publication and will be formally accepted for publication once it meets all outstanding technical requirements.

Kind regards,

Andrew Halford

Academic Editor

PLOS ONE

Additional Editor Comments (optional):

I note the authors have taken on board all the reviewers comments and in so doing have improved the manuscript significantly. Analyses of the impact of COVID on Pacific Island communities are very much needed so this contribution is welcomed.
---

## [Editor Report · Acceptance letter]

18 Apr 2023

PONE-D-22-15433R1 

Roadside sales activities in a South Pacific Island (Bora-Bora) reveal sustainable strategies for local food supply during a pandemic 

Dear Dr. Bertucci:

I'm pleased to inform you that your manuscript has been deemed suitable for publication in PLOS ONE. Congratulations! Your manuscript is now with our production department. 

Kind regards, 

on behalf of

Dr. Andrew Halford 

Academic Editor

PLOS ONE